# Degradation of Structurally Modified Polylactide under the Controlled Composting of Food Waste

**DOI:** 10.3390/polym15194017

**Published:** 2023-10-07

**Authors:** Elena Trofimchuk, Valeria Ostrikova, Olga Ivanova, Marina Moskvina, Anna Plutalova, Tatyana Grokhovskaya, Anna Shchelushkina, Alexander Efimov, Elena Chernikova, Shenghua Zhang, Vladimir Mironov

**Affiliations:** 1Department of Chemistry, Moscow State University, Moscow 119991, Russia; olga12hf@gmail.com (O.I.); moskvina203@yandex.ru (M.M.); annaplutalova@gmail.com (A.P.); groch@genebee.msu.ru (T.G.); efimov@belozersky.msu.ru (A.E.); chernikova_elena@mail.ru (E.C.); 2Scientific Laboratory “Advanced Composite Materials and Technologies”, Plekhanov Russian University of Economics, Moscow 117997, Russia; 3Winogradsky Institute of Microbiology, Federal Research Center of Biotechnology, Russian Academy of Sciences, Moscow 119071, Russia; v-ostrikova@mail.ru (V.O.); schyolushkina@yandex.ru (A.S.); 7390530@gmail.com (V.M.); 4College of Harbour and Coastal Engineering, Jimei University, Xiamen 361021, China; shzhang@jmu.edu.cn

**Keywords:** polylactide, composting, hydrolytic degradation, biodegradation, structure, gel permeation chromatography

## Abstract

The degradation of polylactide (PLA) films of different structures under conditions of controlled composting has been studied. We have demonstrated that PLA underwent degradation within one month in a substrate that simulated standard industrial composting. Regardless of the initial structure of the samples, the number-average molecular weight (M_n_) decreased to 4 kDa while the degree of crystallinity increased to about 70% after 21 days of composting. Addition of an inoculant to the standard substrate resulted in the accelerated degradation of the PLA samples for one week due to an abiotic hydrolysis. These findings have confirmed that industrial composting could solve the problem of plastic disposal at least for PLA.

## 1. Introduction

Due to the growing environmental problems associated with the accumulation of the waste from hardly degradable plastics, their replacement with biodegradable polymers seems genuine. Among other biodegradable polymers, polylactide (PLA) which is a thermoplastic aliphatic polyester synthesized from plant sources [1,2] can be considered a promising candidate for such a substitution. According to the prediction of [3], the concentration of PLA in the food waste stream may reach 8–10 wt.% by the year 2030.

Amorphous PLA, which is often used in the manufacture of packaging, has several advantages, such as high modulus and strength, processability, and non-toxicity. However, it also has drawbacks, which include a low thermal stability and softening temperature (glass transition temperature approx. 60 °C) [4], low impact strength (less than 5 kJ m^−2^) [5], and increased tensile fragility (elongation at break often does not exceed 10–30%) [6]. The known approaches to produce PLA with high performance properties are based on (1) the copolymerization of lactide with various monomers [7,8]; (2) the addition of plasticizers [9,10]; (3) mixing PLA with other polymers or particles of inorganic nature [11,12,13]; and (4) using structural and mechanical approaches [14,15,16,17]. The latter can include orientational strengthening and crystallization, which are easy to perform and do not require any special and expensive synthesis, the use of organic solvents or compatibilizers. Another drawback of PLA is its rather long degradation time (from 2 to 5 years) [18,19] both from waste [20] or landfill disposal [21,22]. Evidently, improving the utilization properties of the polymer may affect its ability to degrade, and thus it requires careful investigation. 

The decomposition of polyesters (including PLA) is known to occur via hydrolysis of ester bonds under the action of water [18,23]. According to Pitt’s theory [24], this reaction is described by third-order kinetics, and its rate depends on the concentration of ester bonds, the amount of absorbed water, and the concentration of terminal carboxyl groups formed as a result of hydrolysis. In solid polymeric materials this reaction is diffusion-controlled, and its rate is largely determined by the rate of diffusion of water molecules into the bulk of the polymer and the reaction products into the environment. The formation of orientational order and crystal structure retards diffusion processes, thereby reducing the initial rate of hydrolytic degradation [25]. The most important factors affecting the rate of polymer decomposition include the degree of crystallinity and the size of the crystallites [26], since the access of water molecules to the denser crystalline regions of the material becomes difficult, and degradation in these regions proceeds much slower than in amorphous regions. Thus, degradation proceeds firstly in the amorphous regions, resulting in an increase in the degree of crystallinity. At this stage, the size of the crystallites does not significantly affect the rate of the process. The degree of crystallinity decreases only when the crystalline regions start degrading [27]. At this stage, the size of the crystallites affects the degradation rate [28]. However, an inverse dependence of the rate of PLA degradation on the degree of crystallinity has been observed in [29]. The degradation of the amorphous regions of PLA results in the accumulation of catalytically active carboxyl groups and causes an autocatalytic acceleration of the degradation of the polymer sample [30].

The industrial composting process allows us to solve the problem of the low rate of PLA decomposition. Composting is an exothermic process of biological oxidation, in which an organic substrate is biodegraded by a mixed population of microorganisms, including bacteria, archaea, and fungi [31]. It is accompanied by a temperature increase in the composted material of up to 50–60 °C and even higher with a moisture content of about 60% at the beginning of the process and 40–50% at the end of compost maturation. Such conditions (a temperature above the glass transition and sufficiently high humidity) are favorable for the hydrolytic degradation of ester bonds [1,32]. It is important to note that both abiotic and biotic factors affect the substrate during composting [33]. Thus, the pH of the waste may decrease from 7 to 4–5 during composting, as a result of the metabolism of lactic acid bacteria, yeasts, molds, and then it increases to 8–9 pH due to the decomposition of organic acids and the release of ammonia [34]. It is believed that microorganisms [35] participate in the degradation process only after the preliminary chemical degradation of synthetic polymers resulting in the decrease in their molecular weight to 10 kDa or less [36]. Biodegradation can also begin with the adhesion of microbial cells to the surface of the PLA material with a subsequent secreting of exoenzymes [37]. When plastics degrade under the action of mixed cultures, the polymer can also be affected by aggressive metabolites (e.g., acids) [38]. It is important to note that PLA decomposition products may change the compost environment and microbial population [33], for example, as a result of lactic acid formation during hydrolysis [39]. In addition, the enzymatic degradation of highly crystalline PLA samples is much slower than that of amorphous samples [36,40]. Thus, the improvement of the physical, mechanical, and consumer properties of PLA, on the one hand, expands the areas of application of this material, and on the other hand, reduces its biodegradability and the possibility of polymer waste recycling, including under industrial composting conditions.

The aim of this research was to study the effect of the structural and mechanical modification of PLA films, achieved by orientation and/or crystallization, on their ability to degrade under conditions of controlled composting as a part of food waste. The degradation process was monitored through the analysis of the molecular weight characteristics and mechanical properties of PLA. Particular attention was paid to the influence of substrate composition and composting dynamics on the abiotic and biotic mechanisms of PLA degradation.

## 2. Materials and Methods

### 2.1. Materials

PLA (NatureWorks LLC, Plymouth, USA) with a content of D–isomer units of 4%, a number-average molecular weight *M*_n_ equal to 76 kDa, and dispersity *Đ*_M_ = 2.2 is used in this research. PLA has glass transition temperature *T*_g_ equal to 63 °C, crystallization temperature *T*_cryst_ = 119 °C, and melting temperature *T*_melt_ = 149 °C. 

Four film samples with different structures have been prepared to study their degradation ability:

**PLA–1** is a transparent, amorphous, nearly isotropic commercial film (NatureWorks, USA) with a thickness *h* of 195 ± 5 μm produced via extrusion with rapid cooling (quenching).

**PLA–2** is produced from **PLA–1** film via cold rolling at room temperature (20–25 °C). Rolling is performed on laboratory rollers in the gap between two rollers rotating at the same speed. In this method, the length of the polymer sample increases in the direction of rolling and, accordingly, its thickness decreases, while its width remains practically unchanged. The degree of rolling λ = 2.1 (equal to the degree of orientation) is evaluated as the ratio of the initial thickness of the PLA film to its thickness after rolling. After rolling, the film retains its transparency, and a periodic stripe pattern appears on it.

**PLA–3** is obtained from **PLA–1** film via isothermal crystallization in ethanol at 55 °C for 45 min, followed by vacuum drying to constant weight. During crystallization, the thickness of the initial amorphous film increases slightly, and the sample becomes opaque and white in color.

**PLA–4** is produced from a **PLA–1** film via orientational stretching at room temperature in a 50% aqueous ethanol medium at a rate of 2.5 mm/min to a tensile strain of 4.5 (this value is equal to the ratio of the final sample length to its initial value). In this case, the development of large deformations occurs according to the crazing mechanism through the formation of a highly dispersed fibrillar–porous structure with its subsequent collapse. As previously shown in [41,42], orientation of the polymer film is followed by “cold” crystallization. The samples were dried under isometric conditions to constant weight after stretching. The resulting samples become opaque and white.

### 2.2. Composting Conditions

In the experiments with composting, two types of substrates are used: standard and accelerated. The standard substrate (standard compost, **SC**) consists of a mixture of food waste (FW). In the accelerated substrate (accelerated compost, **AC**), a solid filler is replaced by inoculum, resulting in intensification of the composting process. Mature compost from a mixture of FW and filler from a previous experiment with a composting period of 56–98 days was used as an inoculum.

The FW, consisting of expired products of typical composition, is obtained at a waste processing facility (Moscow region, coordinates: 56.048547, 36.996277) (wt.%): potato, 18.0; cabbage, 18.0; apple, 7.2; orange, 7.2; banana, 7.2; minced meat, 3.6; fish, 1.4; bread, 7.2; quark, 1.4; and chicken egg, 0.7.

The composition of the solid filler includes (wt.%) waste paper (napkins, packaging, newspapers, magazines), 11.0; plastic (polyethylene terephthalate (PET), polyethylene (PE), Tetra Pak^®^, Hong Kong, China), 10.6; textiles (nylon, cotton, wool), 0.7; and wood (chips 10–20 mm), 5.8.

Each substrate is mixed in the following ratio: 71.9 wt.% FW and 28.1 wt.% solid filler or inoculum. Preliminarily, the components of the FW and the solid filler were crushed to a size of 10–20 mm and thoroughly mixed. SC was frozen at –20 °C after preparation and thawed under normal conditions for 1 day before the experiment. The experiment with AC was carried out immediately after preparation.

The types of substrates for SC and AC were chosen as they are the most common in the technology for the industrial composting of SC and the organic fraction of municipal solid waste in Russia. The freezing stage of the SC simulated the climatic conditions of the Arctic. The physicochemical parameters of the original substrates are shown in Table 1.

The biodegradation of PLA under composting conditions was carried out during a 21-day pilot test in accordance with the ISO 16929:2021 standard on a laboratory test bench as described previously [34] in the group of microbial processes for the conversion of organic waste (Research Center of Biotechnology RAS, Moscow, Russian Federation). The working volume of the reaction chamber was 10 dm^3^; 2 chambers were used simultaneously for each type of substrate (two parallel experiments). For the experiment, 3 PLA samples of each type were mixed with the substrate and placed in 50 mL perforated Falcon type plastic tubes, which were placed in the reaction chamber containing the substrate as shown in Figure 1.

PLA samples of each type were taken from each tube after 7, 14, and 21 days for the SC substrate and after 4 and 6 days for the AC substrate. Sampling times to complete degradation were determined during preliminary experiments. After sampling, each sample was carefully washed with distilled water, dried separately in an open-lid Petri dish for 24 h in air, and then stored in an individual sealed plastic bag at room temperature.

The substrate was aerated with air at room temperature 24.4 ± 2.7 °C in a volume of 0.156 L min^−1^ kg^−1^ of organic matter (0.156 L min^−1^ kg^−1^ OM) using a blower and rotameter GUZ—2 RMS—A—0.035 (Pribor–M, Moscow, Russia). The aeration rate was chosen based on the optimum value for simultaneous control of ammonia and the volatile sulfur compounds released during waste composting in aerated systems [43,44].

The gas composition of the air passing through the volume of compost was determined daily using a gas analyzer MAG—6 S—1 (Axis, Russia): O_2_ (from 0 to 30.0 ± 0.4 vol%), CO_2_ (from 0 to 10.0 ± 0.1 vol%), NH_3_ (from 0 to 20 ± 4 mg m^−3^), H_2_S (from 0 to 10 ± 2 mg m^−3^), and CH_4_ (from 0 to 5.0 ± 0.2 vol%). 

The temperature in each chamber during the experiment was maintained at the level of the current substrate self-heating temperature ±0.2 °C by means of a heating element and an IVTM—7/2S temperature controller (Axis, Russia).

The pH and electrical conductivity (EC) of the substrate were evaluated using a laboratory analyzer ANION 4150 (Ifraspak–Analit, Novosibirsk, Russia) in a suspension of an aqueous extract of the substrate (10 g of the substrate was placed in 300 mL of distilled water [45]) and in a condensed liquid obtained from the air after compost aeration (exhaust air). For this purpose, air was removed from the reaction chambers through glass flasks, where moisture from it condensed under normal conditions. The condensate was collected for pH and EC measurements and then returned to the substrate to balance the mass fraction of moisture in the range of 50–70 wt.%. The mass fraction of moisture in the substrate was determined via the thermogravimetric method on an Evlas–2M moisture analyzer (Sibagropribor, Novosibirsk, Russia).

The content of organic matter (OM) was determined via the thermogravimetric method at 430 °C [46] in a PM–16M–1200 muffle furnace (EVS, Russia). The total content of organic carbon (C, %) was estimated by dividing the OM by a factor of 1.8 [47]. Total Kjeldahl nitrogen (N, %) in dry matter was determined via sample mineralization in an automatic digester DKL 6 (Velp Scientifica, Usmate Velate, Italy) under heating with concentrated sulfuric acid to 420 °C in the presence of hydrogen peroxide and a mixed catalyst followed by ammonium distillation into a boric acid solution on a semi-automatic distillation unit UDK 139 (Velp Scientifica, Italy) and titration with hydrochloric acid on an Easy Plus pH titrator (Metler Toledo, Greifensee, Switzerland). The C/N ratio was calculated from experimental data.

### 2.3. Characterization of PLA Samples

#### 2.3.1. Gel Permeation Chromatography (GPC)

The apparent molecular weight characteristics of the polymers were determined via GPC on a 1260 Infinity II GPC/SEC Multidetector System chromatograph (Agilent, Santa Clara, CA, USA) equipped with two PLgel 5 μm MIXED B columns (M = 5 × 10^2^ − 1 × 10^7^), at 40 ℃ in THF at a flow rate of 1 mL min^–1^. Average molecular weights were calculated using narrowly dispersed standards of poly(methyl methacrylate). 

#### 2.3.2. Differential Scanning Calorimetry (DSC)

The thermal properties of polymer samples were studied via DSC with a Metler TA4000 instrument (DSC–20 cell) (Metler Toledo, Switzerland) in the temperature range 25–200 °C at a heating rate of 10 °C min^–1^ in a nitrogen atmosphere.

The degree of crystallinity (α) of the samples was determined using the equation
(1)$$\alpha =\frac{\Delta H_{\mathrm{PLA}}-\Delta H_{\mathrm{cryst}}}{\Delta H_{\mathrm{PLA}}\left(100\%\right)}$$
where ∆H_PLA_ is the enthalpy of melting; ∆H_cryst_ is the enthalpy of crystallization; and ∆H_PLA_(100%) is the melting enthalpy of a polymer with a degree of crystallinity of 100% (for PLA ∆H_PLA_(100%) = 93 J g^–1^ [48]).

#### 2.3.3. Determination of Shape Stability

The shape stability of the samples was determined through the change of their geometric shape during heating in air in a SNOL 58/350 IP20 oven (SNOL, Utena, Lithuania). For this purpose, samples of a circular form with 15 mm diameter were cut and placed between glasses to avoid distortion. Then, they were heated to different temperatures (25, 45, 55, 65, 80, and 100 °C) for 15 min, and the changes in their diameter and thickness were measured. The data were presented as the dependence on the ratio of the final diameter (*D*_fin_) of the sample after heating (the side axis of ellipse was measured) to its initial value (*D*_o_) vs. temperature.

#### 2.3.4. X-ray Diffraction Method (XRD)

The phase composition of the samples was studied via XRD on a DRON–3M instrument (Bourevestnik, Saint Petersburg, Russia) with a Si(111) monochromator according to the Θ–2Θ method; X-ray source CuKα (Ni–filter) with a wavelength of 0.154 nm. Preliminary, the films were fixed in special frame holders. The size of the PLA crystallites was determined from the coherent scattering region using the Scherrer equation:(2)$${\;d\;¯}_{\mathrm{cryst}}=\frac{k\lambda }{\beta \cos\theta },$$
where *k* = 0.9 is the shape factor, λ is the X-ray length, β is the full width at half maximum (FWHM) of the most intense reflection corrected for instrumental distortions ($\beta =\sqrt{\beta_{\exp}^{2}-\beta_{0}^{2}}$, where β_exp_ and β_0_ are the experimental and instrumental half-widths of the diffraction maxima, respectively), and θ is the Bragg reflection angle at maximum intensity.

#### 2.3.5. Scanning Electron Microscopy (SEM)

The morphology of PLA samples was studied via SEM on surfaces and splits prepared by brittle fracture technique in liquid nitrogen. The samples were attached to a microscope holder with double-sided carbon adhesive tape, and a 25 nm thick gold layer was deposited using an IB–3 Ion Coater (Eiko, Tokyo, Japan). The samples were examined at the UNIQEM Collection Center for Collective Use of the Federal Research Center for Biotechnology, Russian Academy of Sciences, on a Jeol JSM–IT200 microscope (JEOL, Tokyo, Japan) at an accelerating voltage of 15 kV.

#### 2.3.6. Stress–Strain Test

The mechanical properties of the PLA samples before and after degradation were investigated using the stress–strain test. The tests were performed on an Instron 4301 universal tensile testing machine (Instron Limited, Wycombe, UK) in air at a rate of 2.5 mm/min. Preliminary dog-bone shaped specimens with a working area of 10 mm (length) × 5.2 mm (width) were cut out from the films. Oriented samples PLA–2 and PLA–4 were tested along the pre-strain.

#### 2.3.7. Statistical Analysis

The value of any parameter was measured at least twice for all types of studies in the work. Statistical processing of the data was carried out according to analysis of variance using a test of least significance and comparing the difference between the means of different groups. The significance test was determined at *p* < 0.05. The data values and error bars on the graphs are presented as the mean ± standard deviation. 

### 2.4. Isolation of Pure Cultures of Microorganisms-Destructors of PLA

Cultures of degrading microorganisms were isolated from PLA–4 films. Previously, samples of this type were subjected to composting in the SC. Samples were taken on the 7th day (current conditions in SC: temperature 69.5 ± 4.9 °C, gas release of carbon dioxide 2.9 ± 0.6 g CO_2_ kg^−1^ OM and ammonia 0.002 ± 0.000 mg NH_3_ kg^−1^ OM, pH value 3.0 ± 0.1 and EC 0.29 ± 0.02 mS cm^−1^). To obtain enrichment cultures, the films were placed on the surface of an agar-rich nutrient medium in Petri dishes (composition of the medium, g L^−1^: tryptone, 5; yeast extract, 2.5; glucose, 1; agar, 15). When isolating a culture of the micromycete, an antibiotic was added to the medium (100 mg L^−1^ tetracycline). The medium was chosen due to its ability to provide for growth of both bacteria and fungi. Pure cultures of microorganisms were obtained using medium of the same composition. Incubation was carried out at 30 °C. Further identification was carried out for the three isolated pure cultures. Sequencing for isolated cultures was performed on the 16S rRNA gene region and the ITS region for bacteria and fungi, respectively. Phylogenetic trees were built using Mega11 [49].

## 3. Results

Orientation and crystallization are used commonly to improve the consumer properties of polymeric materials. Orientation usually results in an increase in the mechanical properties of polymers, i.e., a decrease in brittleness and an increase in strength. Crystallization leads to an increase in the stiffness and shape stability of the material. The PLA samples used in this work differ in their degree of crystallinity and orientation order. Their characteristics and appearance are presented in Table 2 and Figure 2, respectively. As expected, the molecular weight characteristics of the PLA samples are similar.

Figure 3 shows the stress–strain curves of the initial PLA samples. The mechanical behavior of the amorphous isotropic film (PLA–1) is typical of PLA. Its stretching in air is accompanied by the formation of crazes and, consequently, the sample becomes opaque. Therefore, the use of such material is possible until it reaches the yielding point, which is observed at a tensile strain of 4% and is equal to 58 MPa, although the value of the breaking elongation can reach 90%.

The crystallization of PLA to relatively low degree of crystallinity (20%) has little effect on the mechanical properties of the polymer. The values of the modulus of elasticity and the yield point remain practically unchanged, while the elongation at break decreases to about 20%. Orientation has a significant effect on the strength properties of PLA. The yield point increases by 50% (to 88 MPa) for an oriented amorphous film (PLA–2) and by more than 80% (to 106 MPa) for an oriented semi-crystalline film (PLA–4). Orientation also leads to an increase in the modulus of elasticity in the longitudinal direction by about 1.5 times, up to 2–2.5 GPa. It is important to note that the tensile mechanism of oriented PLA samples changes, i.e., their deformation proceeds uniformly, with a gradual narrowing of the working part, and the crazing process is suppressed.

Figure 4 illustrates the shape stability of PLA samples after heating at various temperatures.

It turns out that the semi-crystalline isotropic PLA–3 sample has the best shape stability, maintaining its geometric dimensions in all directions, including thickness, even at 100 °C. Other samples begin to shrink rapidly around the glass transition temperature (55–65 °C). For example, the PLA–1 film is compressed equally in all directions in a plane, which leads to the preservation of its original round shape and a 20% decrease in diameter. The oriented samples PLA–2 and PLA–4 shrink in the direction opposite to the pre–orientation and take on an oval shape. The PLA–2 sample demonstrates the largest change in diameter, which is reduced by 2.3 times. Simultaneously, the thickness of the samples increases by about 1.3 times for PLA–1 and PLA–4, and by 2.7 times for PLA–2.

Thus, all PLA samples exhibit different mechanical properties and heat resistance. 

### 3.1. Composting of PLA in the SC Substrate

It is known [34] that composting goes through several stages. The *lag phase* (retarded growth of microorganisms) is characterized by temperatures below 40 °C. The *mesophilic phase* is due to the activity of lactic acid microorganisms and the accumulation of organic acids; it starts when the temperature rises to 45 °C and the pH value drops to 4–5. The *thermophilic phase* is accompanied by an active growth of microorganisms and release of ammonia; it is observed when the temperature increases to 65–70 °C, and the pH to 8–9. The *maturation phase* begins, when a complex of humic acids is finally formed. At this final stage, the activity of the microorganisms begins to decrease due to the depletion of food resources, and the temperature slowly drops to the values of the mesophilic phase.

Figure 5 (dashed lines) presents the dependence of temperature, pH, EC, and CO_2_ and NH_3_ content in the SC substrate on the process time, which characterize the dynamics of PLA composting.

The temperature of the substrate rises from 13.4 ± 0.4 to 42.5 ± 2.0 °C (Figure 5A) on the first day due to the release of heat owing to the vital activity of microorganisms. The latter is accompanied by the formation of 0.52 ± 0.23 g CO_2_ kg^–1^ OM (Figure 5C) and 0.086 mg NH_3_ kg^–1^ OM (Figure 5D), confirming the decomposition of the organic substances of FW including protein compounds. Then, the temperature gradually increases to its maximum value of 71.9 ± 2.2 °C for 7 days. During this period, the lactic acid fermentation of carbohydrates occurs, resulting in the formation of organic acids and a decrease in pH to 2.8 ± 0.4 and EC to 0.29 ± 0.01 mS cm^–1^ (Figure 5B). These parameters correspond to the mesophilic stage of composting. In Figure 5C, an increase in CO_2_ release can be seen, reaching a local maximum of 3.5 ± 0.6 g CO_2_ kg^−1^ OM, supporting the active decomposition of the OM. After 10 days, a significant decrease in temperature (up to 54.0 ± 5.3 °C) and oxygen consumption is observed, which can be associated with an intensive decrease in the mass fraction of moisture in the substrate due to evaporation, and thus the microorganisms become less active. The addition of water condensate to the substrate leads to the recovery of microbial activity, and the temperature and CO_2_ release are increased to a maximum of 72.7 ± 4.5 °C and 4.3 ± 0.4 g CO_2_ kg^−1^ OM for 15 days, respectively. This period (Figure 5B) is also accompanied by a sharp increase in pH (8.4 ± 0.9) and electrical conductivity (5.80 ± 0.29 mS cm^–1^) due to the consumption of organic acids by microorganisms and the formation of ammonium compounds during the decomposition of FW proteins. Such conditions cause the emission of ammonia, the concentration of which in the exhaust air was detected at a maximum level of 0.14 ± 0.04 mg NH_3_ kg^−1^ OM on the 18th day. In total, during the SC composting period of 21 days, the average temperature was 60.3 ± 3.9 °C, the total release of carbon dioxide was 49.4 ± 6.9 g CO_2_ kg^−1^ OM and ammonia 0.72 ± 0.14 mg NH_3_ kg^−1^ OM. During the experiment, methane and hydrogen sulfide were also detected in the exhaust air in concentrations ranging from 0.15 to 0.35 vol.% and from 0.2 CH_4_ to 1.5 H_2_S mg m^−3^, respectively.

Figure 6 shows the appearance of the PLA samples after different periods of their exposure to the SC. After the first week (corresponding to the end of the mesophilic stage and the beginning of the thermophilic stage), all the films keep their integrity, but PLA–1 and PLA–2 loose transparency and become white and opaque. It may be caused by the crystallization of the polymer due to the increase in the substrate temperature above the glass transition temperature of the PLA. The stress–strain test shows that only the PLA–4 sample retains its mechanical strength, although its breaking points are significantly reduced (stress about 20 MPa and stain 7%). Other PLA samples become very fragile.

After 14 days, all the samples have lost their strength and fragmented into pieces of various shapes. For example, the fragmentation of PLA–2 occurs mainly in the direction of preliminary orientation. By the 21st day, the size of the fragments is reduced significantly, and the films have a brown tint. It is important to note that complete degradation of PLA–2 sample was detected.

Figure 7 presents SEM micrographs of the surfaces and brittle cleavages of the initial PLA samples and after 7 and 14 days of their composting in SC. It can be seen that the cleavages of PLA–2 and PLA–4 have a striped unidirectional structure, which is related to their orientation during their preparation. Pores are detected in the volume of PLA–3, which could be formed during non-equilibrium crystallization in a slightly plasticizing liquid medium. After 7 days, pores of micron and submicron size appear throughout the volume of all samples; pore concentration increases significantly for PLA–3. The intensive pore formation confirms the bulk hydrolytic degradation. The intense pore formation in the PLA-3 sample may be due to the autocatalytic nature of the process localized in the amorphous regions of the polymer. Few morphological changes are observed in the PLA–4 sample, which retains its mechanical strength. Apparently, the simultaneous orientational ordering and the presence of a crystalline phase significantly reduce the rate of water diffusion into the bulk of the material and leads to a decrease in the rate of hydrolysis. At the end of the mesophilic phase (up to the 7th day), only microbial traces are observed on the surfaces of the samples. For example, a residue resembling part of a biofilm is found on PLA–1, and elongated formations resembling bacteria are visible on PLA–4. It can be assumed that during the first week, the degradation of PLA proceeds mainly according to the mechanism of abiotic hydrolysis under elevated temperature and humidity. 

After 14 days of composting (period of the thermophilic phase), various microorganisms on the surface and/or in the volume of all samples are found in the form of rods and spheres, lemon-shaped spores and colonies, as well as biofilms. This may indicate the involvement of the microbiota in the degradation of polymer, which becomes a substrate for microorganisms. Moreover, the degradation is facilitated by the formation of pores on the surface and throughout the volume of the polymer material at the stage of abiotic hydrolysis. Thus, the biodegradation of PLA under SC conditions occurs mainly after abiotic hydrolysis, at the thermophilic phase.

An important issue concerns the change in the structure of PLA samples during composting. As noted earlier, when amorphous samples of PLA–1 and PLA–2 were treated with SC, they became white, which may indicate the crystallization process. The changes in the degree of crystallinity of the samples were studied via DSC. Figure 8 shows the changes in the shape of the DSC curves for amorphous (e.g., PLA–1) and semi-crystalline (e.g., PLA–4) samples during decomposition. Table 3 shows the values of the degree of crystallinity (α) and the melting temperature of all PLA samples at different periods of composting.

For amorphous PLA–1, a glass transition is observed at 63 °C, a “cold” crystallization peak appears at 119 °C with an enthalpy of about 23 J g^−1^, and a melting point at 149 °C with an enthalpy close to 23 J g^−1^. The presence of an asymmetric endothermic peak at *T*_g_ indicates a reduced enthalpy in PLA due to its physical (structural) aging as a result of a possible improvement in the molecular packing of the polymer chains or an increase in the concentration of low-energy conformations. Under composting conditions (for 7 days) at temperatures above 60 °C, PLA–1 crystallizes; therefore the “cold” crystallization peak is not detected on the DSC curve and the degree of crystallinity is increased to 39% (enthalpy of 37 J g^−1^). After 14–21 days, the degree of crystallinity of PLA already increases to 74% (enthalpy of 69 J g^−1^), and the melting point shifts towards lower temperatures to about 139 °C. These facts can be explained by the facilitation of PLA crystallization due to a decrease in its molecular weight during composting resulting from an abiotic hydrolysis of ester bonds or from biodegradation under the action of the microbiota, mainly in the amorphous regions. The formation of the crystal structure of PLA during hydrolytic degradation was also reported in [50]. For the PLA–2 sample, the same changes in the DSC thermograms are observed as for PLA–1. The difference for the PLA–3 and PLA–4 samples, which have a semi-crystalline structure, is that there are no thermal effects on the DSC curves at the glass transition and “cold” crystallization temperatures, which is typical for semi-crystalline polymers. Composting of the PLA–3 and PLA–4 samples in SC is also accompanied by a gradual increase in the degree of crystallinity to 65–74% and a decrease in the melting point to 145 °C after 14 days. Similar changes of semi-crystalline PLA during its hydrolytic degradation have been described elsewhere [28,51]. It is important to note that a slight decrease in the degree of crystallinity of PLA–4 was observed after 21 days (Table 3), which may be associated with the destruction of crystallites due to the occurrence of degradation processes in the final stages. The degradation temperature of PLA–4 in the DSC cell is significantly reduced (to 175 °C).

X-ray diffraction was used to study the change in the crystal structure of PLA during composting. X-ray diffraction patterns of the original PLA–4 sample and after 7 and 14 days of treatment in SC are presented in Figure 9.

The initial sample is characterized by a so-called metastable α’–crystal structure, for which usually only two or three broadened main diffraction peaks can be identified in the diffraction patterns. In Figure 9A, these are peaks at 16.8° and 22.5°, related to the (200/110) and (210) planes. After 14 days of composting, the crystal structure of PLA improves, the crystal reflections become more pronounced and narrower. Probably, a more stable α–crystalline phase has formed [52], which is characterized by the appearance of additional diffraction reflections at 14.9° and 19.2° corresponding to the planes (010) and (203). Calculations of crystallite sizes from the most intense peak (200/110) using the Scherrer formula show that the crystallite size is 8 nm in the initial PLA–4 sample, increasing to 12 nm after 7 days of composting and to 17 nm after 14 days of composting. Since the degree of crystallinity increases simultaneously with the growth in crystallite size (Table 3), it can be assumed that the growth of primary crystallites and the improvement of the crystal structure occur during degradation.

Previously, it was assumed that the structural and morphological changes in PLA samples were associated with a decrease in the molecular weight of macromolecules due to degradation under composting conditions. Figure 10 shows changes in the molecular weight distribution (MWD) of the PLA samples before and after composting. It can be seen that PLA decomposition in SC is accompanied by an intense decrease in the molecular weight of the polymer. The similar changes of the MWD curves are observed for all PLA samples.

During the first week, *M*_n_ of all PLA samples decreases by 2–3 times except PLA–4, which has an oriented crystal structure (Figure 11). This finding is in good agreement with both the SEM data (Figure 7) and the results of the mechanical stress–strain tests. After 14 days of composting, the structure of all PLA samples becomes similar due to shrinkage (Figure 4) and crystallization (Table 3) above the glass transition temperature. Therefore, changes in the molecular weight characteristics of structurally similar PLA samples are close over this period: the values of *M*_n_ fall to about 7 kDa and *Đ*_M_ to 1.5. By the 21st day, the decrease in the values of *M*_n_ and *Đ*_M_ continues and reaches 4 kDa and 1.1, respectively.

Thus, the degradation of PLA is accompanied by a decrease in the molecular weight of the polymer to 4 kDa in 21 days, and the complete disappearance of the film samples takes place in 1 month. The degradation occurs regardless of the presence of an oriented and/or crystalline structure in the polymer sample under conditions simulating standard industrial composting (it is characterized by the gradual self-heating of the substrate to a temperature of 65–72 °C over 1 week, high humidity, and the presence of various microorganisms). However, the question of the role of abiotic hydrolysis and biodegradation in the overall process of PLA degradation remains unresolved. 

### 3.2. Composting of PLA in the AC Substrate

In order to understand the influence of different mechanisms on PLA degradation, an experiment was conducted with a modified substrate composition (AC), in which the inert filler was replaced by an inoculum already containing a microbial community capable of rapid multiplication under favorable conditions (presence of a nutrient medium and moisture). Since the decomposition of structurally different PLA samples under SC conditions are similar, isotropic amorphous PLA–1 and oriented semi-crystalline PLA–4 were chosen for further investigations. Figure 5 (solid lines) presents the dependencies of different parameters of the AC substrate on time, showing the dynamics of composting PLA as a part of the AC substrate. There are noticeable differences in the conditions between SC and AC due to the addition of inoculum to the latter. The intensification of the biodegradation processes occurs in AC, so that a rapid increase in temperature is observed. Thus, there is a short-term mesophilic stage (12–16 h) with a temperature of 13–42 °C, then another rapid increase in the temperature of the substrate to 72.0 ± 2.0 °C on the 2nd day and a maximum temperature of 72.9 ± 2.2 °C is reached on the 5th day (Figure 5A). The release of carbon dioxide during this period rises from 1.5 ± 0.1 to 3.5 ± 0.6 g CO_2_ kg^–1^ OM, and the change in pH rises from 7.0 ± 0.6 to 7.8 ± 0.7. It can be noted that the decrease in pH, which is characteristic of SC, does not occur in AC composting (Figure 5B). The high temperature probably leads to the death of most of the mesophilic microorganisms and to a decrease in the activity of the thermotolerant (resistant in the temperature range of 20–60 °C) compost microbiota. Thus, on 7th day ammonia starts to release (Figure 5D) at a concentration of 0.38 ± 0.07 mg NH_3_ kg^−1^ OM, which causes an increase in pH to 8.7 ± 0.6 and EC to 5.1 ± 0.4 mS cm^−1^ (Figure 5B). The maximum concentration of ammonia (11.1 ± 0.1 mg NH_3_ kg^−1^ OM) is observed on days 12–14, which is associated with the decomposition of FW protein compounds. The values of pH (9.5 ± 0.4) and EC (10.5 ± 0.9 mS cm^−1^) are also at a maximum on the 12th day. The temperature did not fall below 57 °C throughout experiment, and its average value over a period of 21 days was 64.2 ± 3.4 °C. The total amount of carbon dioxide generated for 21 days was 47.6 ± 6.4 g CO_2_ kg^−1^ OM, and ammonia was8.8 ± 1.3 mg NH_3_ kg^−1^ OM. Thus, a higher temperature and pH are observed in the substrate during AC composting during the first week, which should have affected the rate of decomposition of the PLA samples. Indeed, fragmentation of the polymer samples was already observed after 4 days of treatment with AC, and PLA–1 and PLA–4 had broken into small pieces by the 6th day (Figure 12). One day later, the samples were no longer found. The morphology of PLA–1 and PLA–4 after 6 days exposure to AC differs significantly from the same samples after 7 days exposure to SC (Figure 7). It can be seen that the surface of the PLA–1 sample is covered with microcracks, a large number of pores about 250 nm in size are found throughout the volume, and the structure of the PLA–1 film looks like a sponge. The morphology of the PLA–4 film after being in the AC for 6 days is similar to that of the film after being in the SC for 14 days. However, it is important to note that there are an insignificant number of microorganisms on the surfaces and note their absence in the volume of the PLA samples treated with active compost.

The changes in the degree of crystallinity and melting point of PLA samples during exposure to AC are shown in Table 3. As can be seen, after 4 days of AC treatment these values (*T*_melt_ = 151–152 °C and α = 48–55%) are comparable with the values determined after 7 days of SC treatment. The changes observed after 6 days of AC treatment are comparable with the changes determined after 14 days of SC treatment. Figure 13 presents the MWD curves of the PLA–1 and PLA–4 samples after their exposure to the AC. Similar changes of MWD are observed for both samples. When comparing data for different composting conditions (Figure 10 and Figure 13), one can conclude that MWD decreases are larger for PLA samples after 4 days of being in AC rather than in SC. After that, the behavior of the samples becomes similar. These results confirm the assumption about the defining role of temperature on the rate of the hydrolytic degradation of PLA.

To identify the key factor influencing the process of the hydrolytic degradation of PLA, a model experiment was conducted. Two parts of the PLA–1 film were heated simultaneously at 65 °C for 6 days. However, one part was heated in air and another part in deionized water (abiotic conditions). After heating in dry air, the molecular weight characteristics are kept practically constant (curves 0 and “6 air” for PLA–1 in Figure 13). This is due to the low rate of hydrolysis of the ester bonds in the absence of water. Heating PLA in water causes a significant decrease in the molecular weight of PLA to *M*_n_ = 5.8 kDa (curves 6 and “6 H_2_O” for PLA–1 in Figure 13). Thus, it may be assumed that the degradation of the PLA samples under AC conditions proceeds mainly as a result of abiotic hydrolysis.

The addition of the inoculant to the compostable substrate prevents its acidification, allowing it to significantly reduce the duration of the mesophilic phase and quickly transfer the composting process to the thermophilic phase, characterized by high temperatures of 65–72 °C (5–12 °C higher than the glass transition temperature of PLA). The decomposition time of PLA under active composting conditions can be reduced to 1 week compared to standard composting, where the degradation process continues for about 1 month. It is important to note that under the conditions of AC, the abiotic hydrolysis dominates; under the conditions of SC, the processes of biodegradation at the thermophilic stage seem to play a significant role.

### 3.3. Identification of Microorganisms–Destructors of PLA

Several different microorganisms were isolated and identified from the surface of the PLA–4 sample, which was the least active under SC degradation conditions. A micromycete belonging to the genus *Aspergillus*, presumably in the section *Nidulantes*, was found [53]. It is closest to the species *A. stellatus* (Genbank accession number OR192945), according to the results of sequencing and phylogenetic analysis (Figure 14).

Fungi of this genus are common in the compost community [34]. It has been shown in [54] that this genus has pronounced thermotolerance among members of the phylum *Ascomycota*. *Aspergillus nidulans* is a source of various enzymes, including lipases, proteases, cutinases, etc. [55]. They are active degradants of a wide range of compounds, and in particular, various plastics [56]. The isolation of a thermotolerant strain of *Aspergillus* sp. capable of degrading polycaprolactone at 50 °C has been reported [57]. These fungi are known to be effective in degrading PLA [35,58,59].

A bacterium belonging to the genus *Lysinibacillus* was also isolated. According to the results of sequencing and phylogenetic analysis, it was similar to the species *L. fusiformis* (Genbank accession number OR192961) (Figure 15). The detected microorganism is a Gram-positive, rod-shaped bacterium.

This species is characterized by thermotolerance, high protease activity [60], and the production of intracellular esterase [61]. Bacteria of this genus are known to have a high potential for the degradation of various compounds. In [62], a *Lysinibacillus* sp. strain capable of degrading polypropylene and polyethylene was isolated; the high activity of a mixture of bacterial cultures of the genus *Lysinibacillus* and fungi of the genus *Aspergillus* in the biodegradation of polyethylene was described in [63]. It is reported [64] that a member of this genus can degrade xylan. Liu et al. [65] found that soil isolates of *Lysinibacillus* can play a significant role in accelerating the degradation of PLA-containing materials.

A bacterium belonging to the genus *Bacillus*, which is closest to the species *B. subtilis* (Genbank accession number OR327054), was isolated from the PLA–4 surface according to the results of sequencing and phylogenetic analysis (Figure 16). It is a species of Gram-positive, spore-forming, aerobic, rod-shaped bacteria. This genus of bacteria is one of the dominant ones in composts [34], producing various enzymes, including amylases, lipases, proteinases, etc. [66], while the phenomenon of thermotolerance is also quite common for them [54], which makes it possible to isolate the culture under the conditions described. Their distinct role in the biodegradation of PLA was first noted in [35,67].

## 4. Conclusions

In summary, the features of the degradation of structurally different PLA films with varying degrees of crystallinity and orientation have been studied under conditions simulating standard industrial composting. It is found that, regardless of the structure of the initial samples, all of them completely decompose (at least they cannot be isolated from the substrate) within 1 month. At the same time, the PLA samples have similar thermal and molecular weight characteristics after 14 days of composting. During the mesophilic phase of the SC, the degradation of PLA appears to proceed predominantly through the mechanism of bulk abiotic hydrolysis with random breaking of ester bonds. In the thermophilic phase, microorganisms are colonized and grow intensively on the surface and in the volume of the polymer films, which may indicate the processes of PLA biodegradation. In particular, three different types of microorganisms that colonized the PLA samples were isolated and identified, namely, a micromycete of the genus *Aspergillus* and two bacteria of the genus *Lysinibacillus* and *Bacillus*, which can produce a wide range of enzymes from the class of hydrolases catalyzing the degradation process of a polymer.

The addition of an inoculum to the original standard substrate has been shown to alter the composting conditions. A significant shortening of the mesophilic phase (less than 1 day) and faster temperature rise to 72 °C are typical for active composting conditions. During active composting, the degradation of PLA samples is significantly accelerated, proceeding mainly through the mechanism of abiotic hydrolysis, and the polymer decomposition is completed in 1 week. This indicates that the main role of the microorganisms in AC is to generate heat.

The results obtained confirm that temperature and humidity are the main factors influencing the rate of the hydrolytic degradation of PLA. Therefore, the conditions of controlled industrial composting seem to be the most optimal for the implementation of a rather fast and predictable approach to the disposal of products made from PLA of various structures, such as packaging and disposable tableware, when plastic is found in the waste along with food residue.

## Figures and Tables

**Figure 1 polymers-15-04017-f001:**
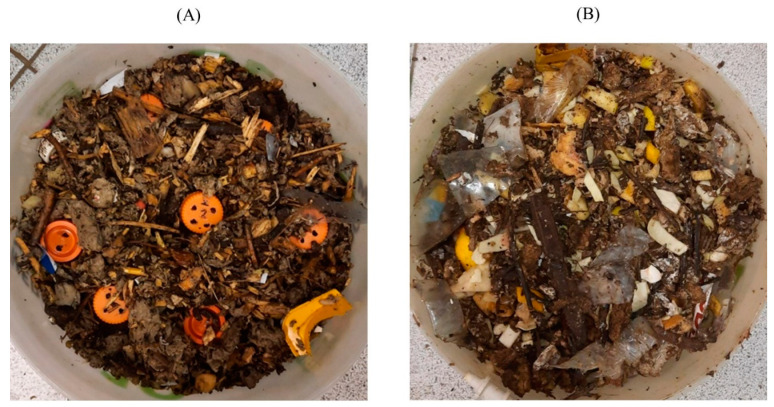
Image of substrates: (**A**)—standard compost, (**B**)—accelerated compost.

**Figure 2 polymers-15-04017-f002:**
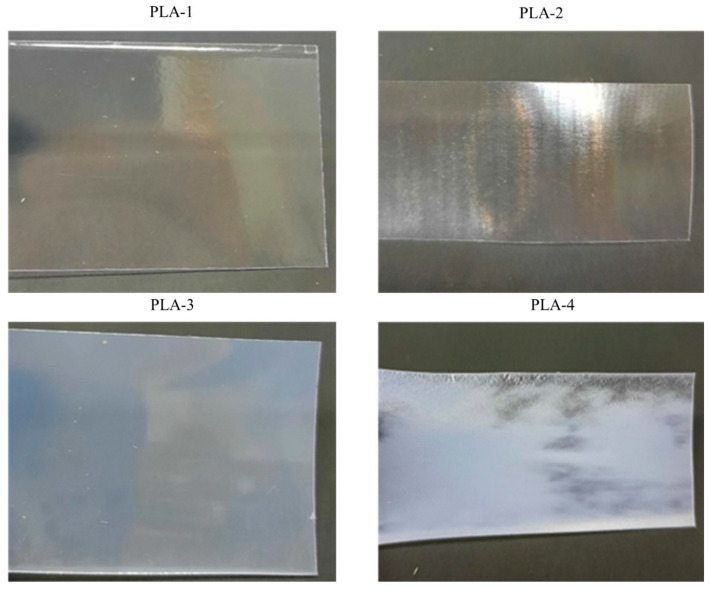
Appearance of initial PLA samples. External view/photos.

**Figure 3 polymers-15-04017-f003:**
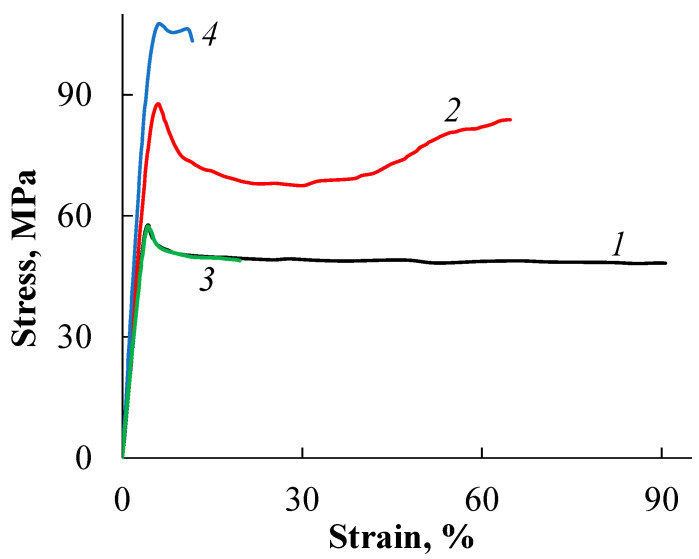
Stress–strain curves of the PLA samples: PLA–1 (1), PLA–2 (2), PLA–3 (3), and PLA–4 (4).

**Figure 4 polymers-15-04017-f004:**
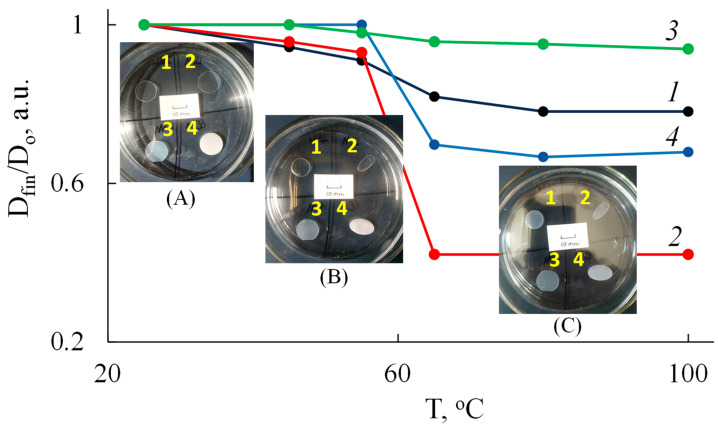
Dependence of the relative change of the diameter of the samples on the temperature for PLA samples: PLA–1 (1), PLA–2 (2), PLA–3 (3), and PLA–4 (4). The insets show photographic images of the samples at (**A**) 25, (**B**) 65, and (**C**) 100 °C.

**Figure 5 polymers-15-04017-f005:**
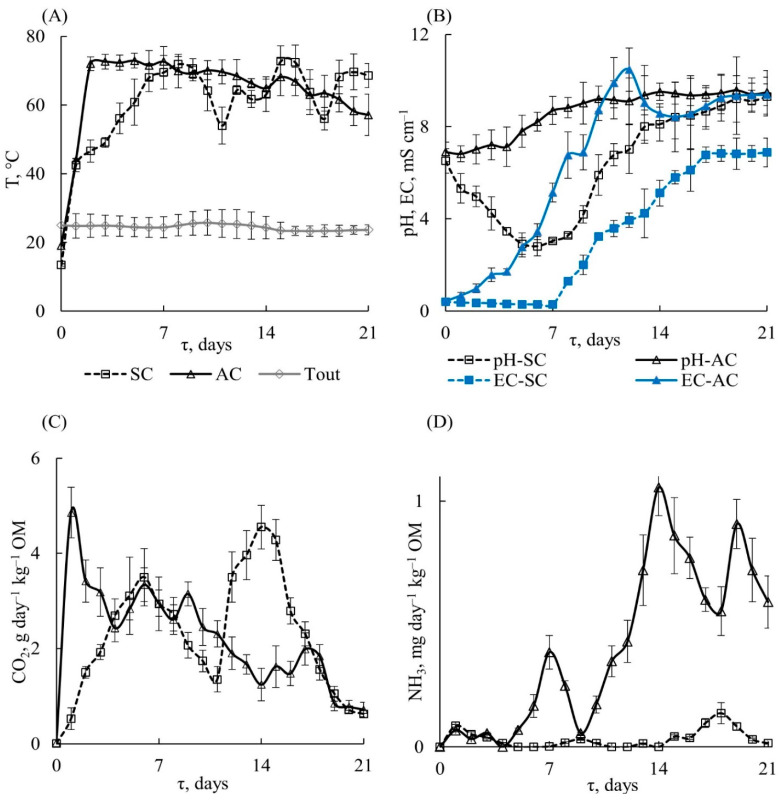
Dependence of (**A**) substrate temperature (T) and outdoor temperature (T_out_), (**B**) substrate pH value and electrical conductivity (EC), (**C**) the content of carbon dioxide CO_2_ and (**D**) ammonia NH_3_ in the exhaust air on the composting time during the composting process in the SC (dashed lines) and AC (solid lines) substrates.

**Figure 6 polymers-15-04017-f006:**
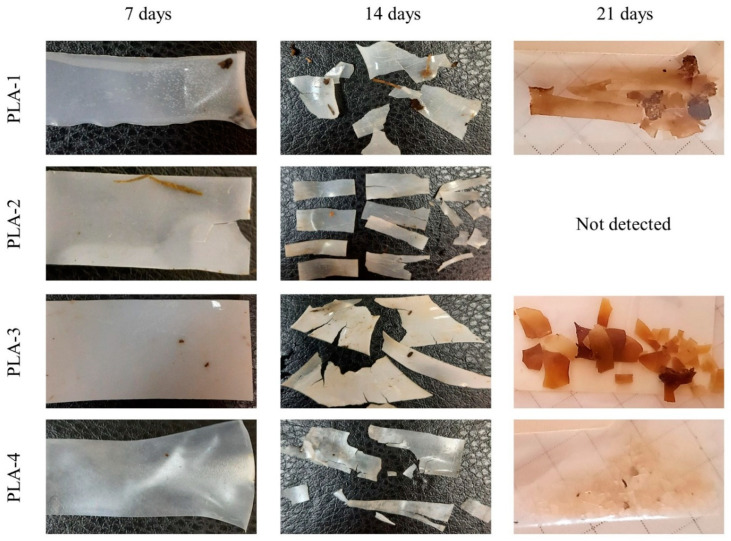
Images of PLA samples (PLA–1, PLA–2, PLA–3, PLA–4) after their exposure to the SC for different times (0, 7, and 14 days).

**Figure 7 polymers-15-04017-f007:**
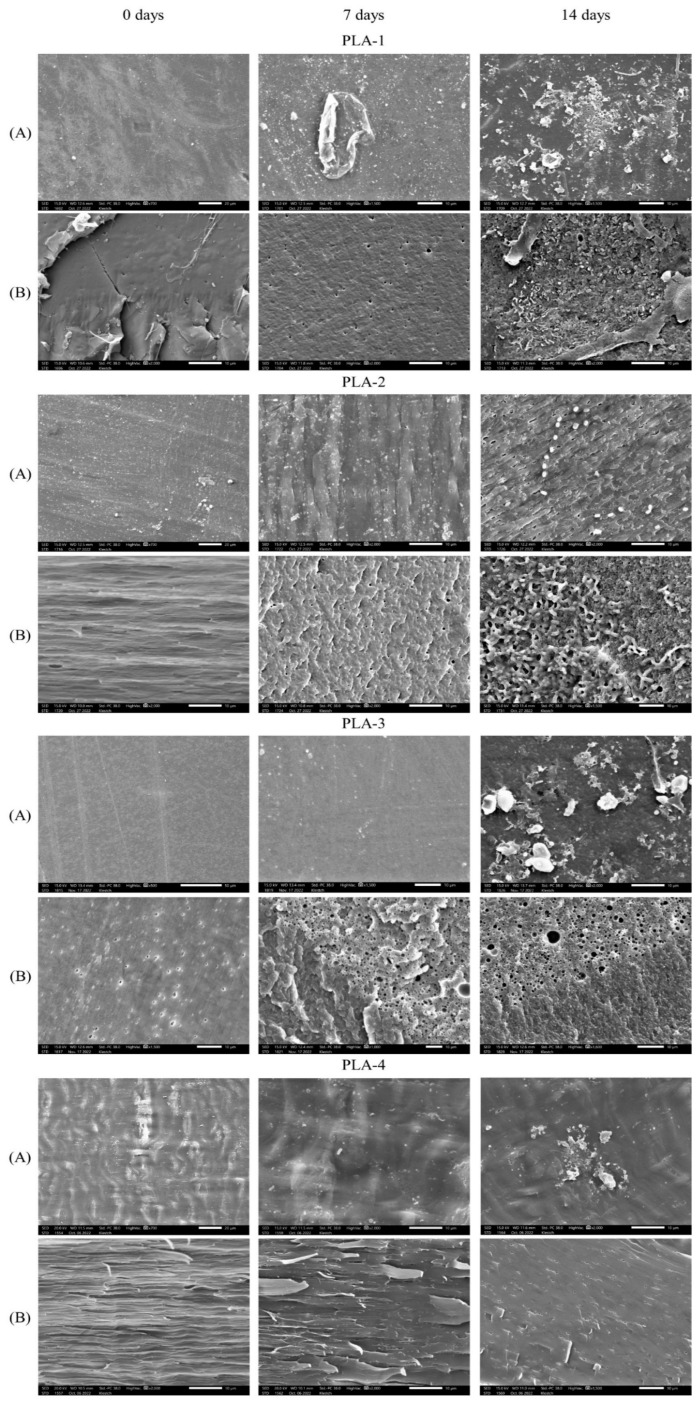
SEM micrographs of (**A**) the surfaces and (**B**) cleavages of the PLA–1, PLA–2, PLA–3, and PLA–4 samples and after their exposure to the SC for 7 and 14 days.

**Figure 8 polymers-15-04017-f008:**
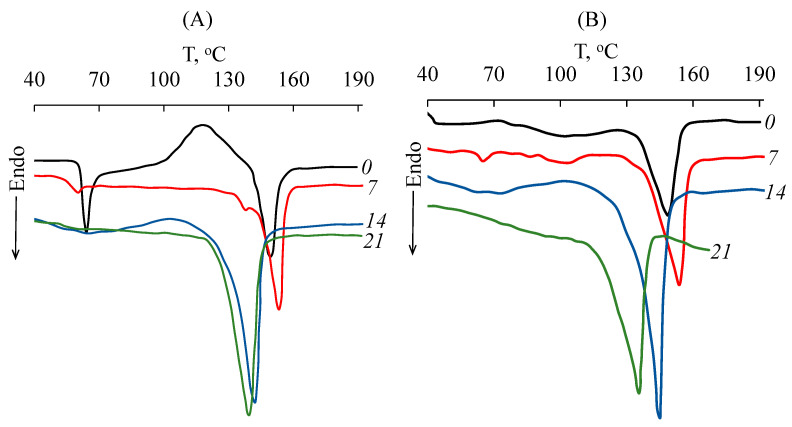
DSC curves of (**A**) PLA–1 and (**B**) PLA–4 samples: 0—initial samples; 7, 14, 21—samples after 7, 14, and 21 days of composting in the SC substrate, respectively.

**Figure 9 polymers-15-04017-f009:**
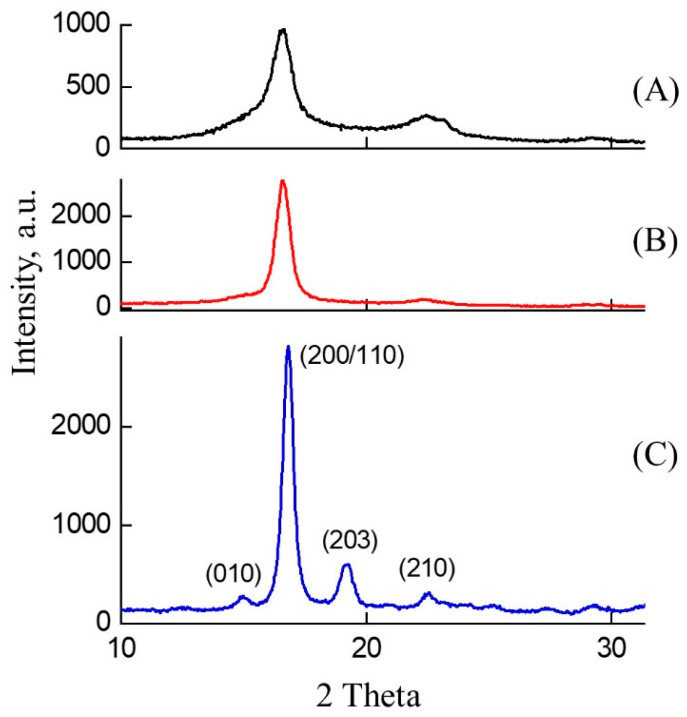
X-ray diffraction patterns of a PLA–4 sample obtained after (**A**) 0, (**B**) 7, and (**C**) 14 days of exposure to the SC. The crystallographic Miller indices are indicated above the reflections.

**Figure 10 polymers-15-04017-f010:**
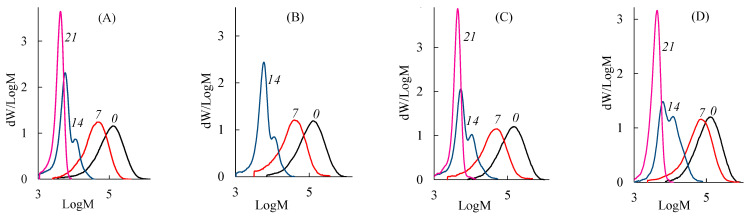
Molecular weight distribution of (**A**) PLA–1, (**B**) PLA–2, (**C**) PLA–3, and (**D**) PLA–4 samples: 0—initial sample; 7, 14, 21—sample after 7, 14, and 21 days of composting in the SC substrate, respectively.

**Figure 11 polymers-15-04017-f011:**
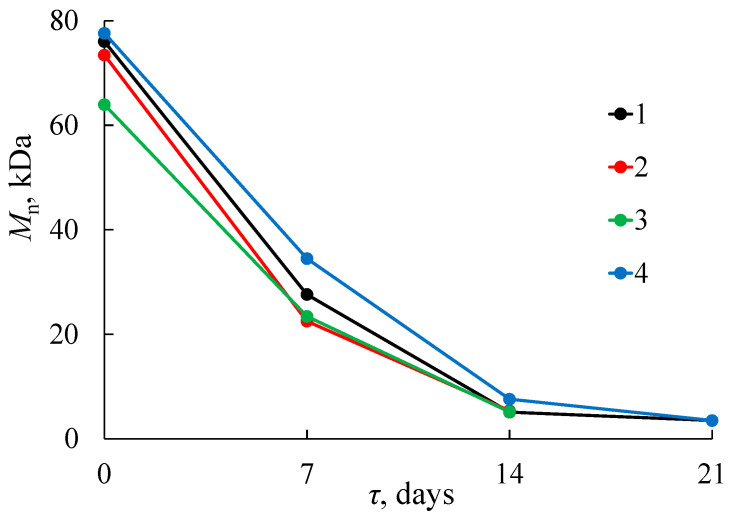
Dependence of *M*_n_ on degradation time τ for PLA–1 (1), PLA–2 (2), PLA–3 (3), and PLA–4 (4) samples after their exposure to the SC.

**Figure 12 polymers-15-04017-f012:**
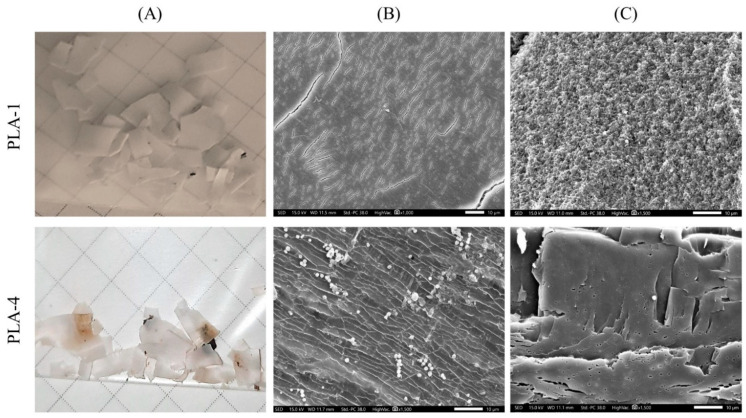
(**A**) Appearance and SEM micrographs of (**B**) the surfaces and (**C**) cleavages of PLA–1 and PLA–4 samples after their exposure to the AC for 6 days.

**Figure 13 polymers-15-04017-f013:**
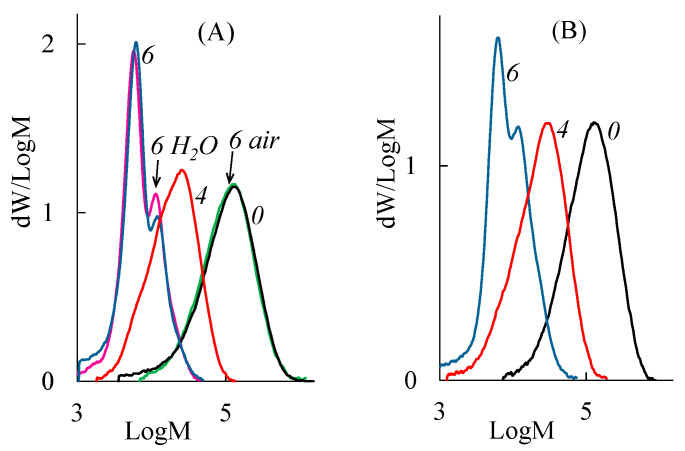
MWD curves of (**A**) PLA–1 and (**B**) PLA–4 samples: 0—initial sample; 4, 6—sample after 4 and 6 days of composting in the AC substrate, respectively; “6 air” and “6 H_2_O”—PLA–1 after 6 days of heating at 65 °C in air and in deionized water, respectively.

**Figure 14 polymers-15-04017-f014:**
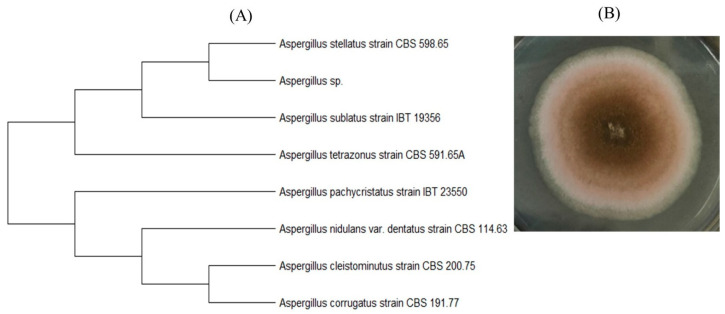
*Aspergillus* sp.: (**A**) position in the phylogenetic tree built on the basis of the region’s ITS sequences; (**B**) micromycete colony grown on agar medium.

**Figure 15 polymers-15-04017-f015:**
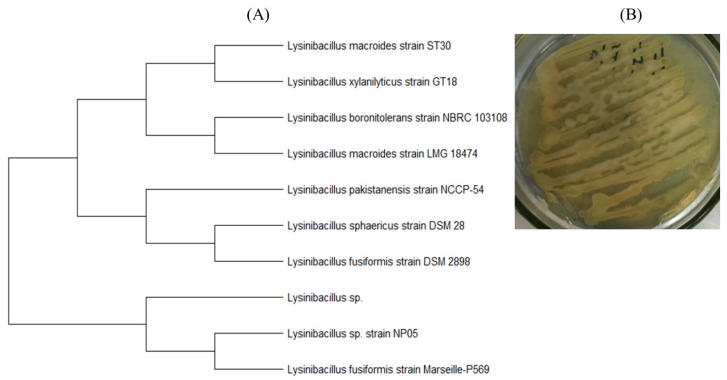
*Lysinibacillus* sp.: (**A**) position in the phylogenetic tree built on the basis of 16S rRNA gene sequences; (**B**) colonies grown on agar medium (streaked).

**Figure 16 polymers-15-04017-f016:**
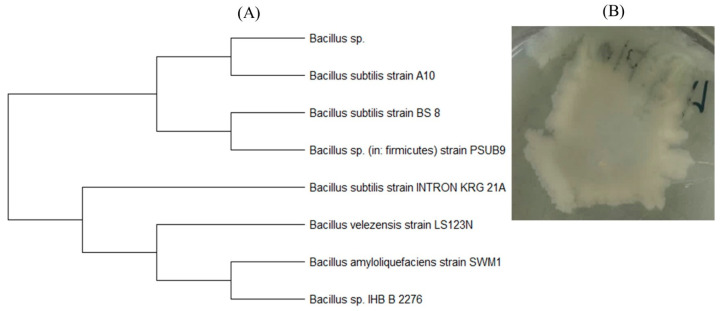
*Bacillus* sp.: (**A**) position in the phylogenetic tree built on the basis of 16S rRNA gene sequences; (**B**) colony grown on agar medium.

**Table 1 polymers-15-04017-t001:** Physicochemical parameters of the used substrates SC and AC.

Parameter	Value *	
	SC	AC
pH	6.5 ± 0.5	6.9 ± 0.4
Electrical conductivity (EC), mS cm^−1^	0.40 ± 0.03	0.42 ± 0.05
Moisture content, wt.%	69.8 ± 1.2	71.4 ± 1.3
C/N	39.5	31.2

*—average value ± standard deviation of three replicates.

**Table 2 polymers-15-04017-t002:** Characteristics of the initial PLA samples.

Type	Sample Description	*h*, µm	α, %	*M*_n_, kDa/*Đ*_M_
PLA–1	Amorphous, isotropic, transparent	195 ± 5	0	76/2.18
PLA–2	Amorphous, oriented (λ = 2.1), transparent	95 ± 3	0	73/1.96
PLA–3	Semi-crystalline, isotropic, white and opaque	215 ± 5	20 ± 3	64/2.07
PLA–4	Semi-crystalline, oriented (λ = 4.5), whitish and opaque	100 ± 7	30 ± 3	78/1.83

**Table 3 polymers-15-04017-t003:** DSC data of PLA samples after composting in SC and AC.

Sample Type	Composting Time, Days	T_melt_, °C	α, %
	**SC**		
PLA–1	0	–	0
	7	153	39
	14	142	74
	21	139	74
PLA–2	0	–	0
	7	154	41
	14	144	71
PLA–3	0	153	20
	7	154	43
	14	144	65
PLA–4	0	149	30
	7	154	47
	14	145	74
	21	135	67
	**AC**		
PLA–1	0	–	0
	4	151	55
	6	144	84
PLA–4	0	149	30
	4	152	48
	6	144	78

## Data Availability

Data is contained within the article.

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
