# Peer review of "Degradation of Structurally Modified Polylactide under the Controlled Composting of Food Waste"

_polymers, 2023, doi:10.3390/polym15194017_

Round 1

Reviewer 1 Report

Authors report the results of their study of the degradation of four morphologically differing PLA samples prepared from the same commercial semicrystalline PLA containing 4% D-units, in two types of composts: inoculated and uninoculated. The conclusion is expected – the PLA disappearance is much faster in inoculated compost. What is perhaps more interesting, it is the observation that the degradation in inoculated compost proceeds roughly with the same speed as purely hydrolytic non-bacterial degradation. This indicates that the main role of bacteria is in the production of heat.

     The article is not written well. It is unnecessarily long, not sufficiently focused on the main thread of the story and really new knowledge. Some parts are trivial, some repeated, etc. Moreover, it contains numerous deficiencies of a basic nature, such as incorrect units of quantities, incorrect or insufficient captions to Figures and Tables, values reported to decimal places and others.

     In addition, if starting purchased PLA exhibits melting and crystallization (see Experimental), one expects semi-crystallinity also in PLA-1 that was prepared by rolling the purchased PLA. I logically suppose that this process was performed at temperature above Tg but below Tm. Semi-crystallinity can be supposed also for PLA-2. However, zero value is reported and details of PLA-1 preparing are not given.   

149     Were the composts and measured samples indeed prepared with the exactness to one ten-thousandth of weight as reported in Experimental.

166.    Table 1.  Not a single unit in the column headed "Units" (which should be in the singular) is correct and/or complete.   

227     Authors could not calculate MW averages from PMMA standards. They determined apparent MW averages related to PMMA standards (See IUPAC Recommendations, Pure Appl. Chem. 2015; 87(1): 71–120, entry 2.13). This fact should be mentioned, but correctly, since overwhelming majority of published GPC MW data for synthetic polymers are based on PS standards.

229     a) What is definition of “normalized weight fraction” ??     
b) Is the SEC coordinate dW/logM written correctly  ??  
c) Note: the symbol w is commonly used for weight fraction.
d) Axis labels do not need to be in Experimental but should be in the figures. But they are missing there.

240     Determination of thermal stability.    Here, it is about a shape stability and not "chemical stability" usually understood under this heading.   Why not to state here that the side axis of ellipse was measured ??

254     CSR   useless abbreviation.

257     FWHM  uspecified.

271     A rather strange sentence.

280     ANOVA   useless abbreviation.

306     Caption to Tab. 2 does not correspond to its content. What chemical property is reported? Quantities not defined.

318     The discussion to Figs 3 and 4 is very curious. These results show nothing more than the textbook knowledge of the effect of crystallinity on the basic mechanical properties of a crystallizable polymer and the shape memory of the deformed but not thermally-relaxed polymer. Discussion does not bring anything new.

323     No crystalline but semi-crystalline films are studied.

348     This paragraph is essentially a repetition of facts from Introduction.

369     Text does not correspond with that to Fig. 5. 

373     Info to Fig. 5 is absolutely insufficient.

425     Fig. 6.  Why low-crystallinity samples survive much longer than those oriented with higher degree of crystallinity ?  Can you reasonably explain it ???  

475     Are data in Tab. 4 in accord with the DSC records in Fig. 8 ???   

Authors report the results of their study of the degradation of four morphologically differing PLA samples prepared from the same commercial semicrystalline PLA containing 4% D-units, in two types of composts: inoculated and uninoculated. The conclusion is expected – the PLA disappearance is much faster in inoculated compost. What is perhaps more interesting, it is the observation that the degradation in inoculated compost proceeds roughly with the same speed as purely hydrolytic non-bacterial degradation. This indicates that the main role of bacteria is in the production of heat.

     The article is not written well. It is unnecessarily long, not sufficiently focused on the main thread of the story and really new knowledge. Some parts are trivial, some repeated, etc. Moreover, it contains numerous deficiencies of a basic nature, such as incorrect units of quantities, incorrect or insufficient captions to Figures and Tables, values reported to decimal places and others.

     In addition, if starting purchased PLA exhibits melting and crystallization (see Experimental), one expects semi-crystallinity also in PLA-1 that was prepared by rolling the purchased PLA. I logically suppose that this process was performed at temperature above Tg but below Tm. Semi-crystallinity can be supposed also for PLA-2. However, zero value is reported and details of PLA-1 preparing are not given.   

149     Were the composts and measured samples indeed prepared with the exactness to one ten-thousandth of weight as reported in Experimental.

166.    Table 1.  Not a single unit in the column headed "Units" (which should be in the singular) is correct and/or complete.   

227     Authors could not calculate MW averages from PMMA standards. They determined apparent MW averages related to PMMA standards (See IUPAC Recommendations, Pure Appl. Chem. 2015; 87(1): 71–120, entry 2.13). This fact should be mentioned, but correctly, since overwhelming majority of published GPC MW data for synthetic polymers are based on PS standards.

229     a) What is definition of “normalized weight fraction” ??     
b) Is the SEC coordinate dW/logM written correctly  ??  
c) Note: the symbol w is commonly used for weight fraction.
d) Axis labels do not need to be in Experimental but should be in the figures. But they are missing there.

240     Determination of thermal stability.    Here, it is about a shape stability and not "chemical stability" usually understood under this heading.   Why not to state here that the side axis of ellipse was measured ??

254     CSR   useless abbreviation.

257     FWHM  uspecified.

271     A rather strange sentence.

280     ANOVA   useless abbreviation.

306     Caption to Tab. 2 does not correspond to its content. What chemical property is reported? Quantities not defined.

318     The discussion to Figs 3 and 4 is very curious. These results show nothing more than the textbook knowledge of the effect of crystallinity on the basic mechanical properties of a crystallizable polymer and the shape memory of the deformed but not thermally-relaxed polymer. Discussion does not bring anything new.

323     No crystalline but semi-crystalline films are studied.

348     This paragraph is essentially a repetition of facts from Introduction.

369     Text does not correspond with that to Fig. 5. 

373     Info to Fig. 5 is absolutely insufficient.

425     Fig. 6.  Why low-crystallinity samples survive much longer than those oriented with higher degree of crystallinity ?  Can you reasonably explain it ???  

475     Are data in Tab. 4 in accord with the DSC records in Fig. 8 ???   

Author Response

Dear Editor! We are grateful to the Reviewer for carefully reading our paper and for the valuable comments and suggestions that have helped to improve the manuscript. We agree with all the comments. We have tried to improve the manuscript significantly and hope that it now meets all requirements. The text of the manuscript has been extensively revised, so only those places where the Reviewer had comments are highlighted in yellow. Best regards, the authors.

 No.

Reviewer Comments

Response to Reviewer 1

1.

The conclusion is expected – the PLA disappearance is much faster in inoculated compost. What is perhaps more interesting, it is the observation that the degradation in inoculated compost proceeds roughly with the same speed as purely hydrolytic non-bacterial degradation. This indicates that the main role of bacteria is in the production of heat.

We would like to thank the reviewer for his opinion. In general, we agree with the conclusion. But it should be noted that standard composting is a common method for which we have studied the mechanism of PLA hydrolysis. The conclusions of the article have been clarified (Lines 655-661).

2.

It (the article) is unnecessarily long, not sufficiently focused on the main thread of the story and really new knowledge. Some parts are trivial, some repeated, etc. Moreover, it contains numerous deficiencies of a basic nature, such as incorrect units of quantities, incorrect or insufficient captions to Figures and Tables, values reported to decimal places and others.

We have tried to correct the text of the article, clarified the captions of the figures (Fig. 5 Lines 342-345, Fig. 10 Lines 492-494, Fig. 13 Lines 572-574), and rounded the experimental values to significant decimal places.

3.

In addition, if starting purchased PLA exhibits melting and crystallization (see Experimental), one expects semi-crystallinity also in PLA-1 that was prepared by rolling the purchased PLA. I logically suppose that this process was performed at temperature above Tg but below Tm. Semi-crystallinity can be supposed also for PLA-2. However, zero value is reported and details of PLA-1 preparing are not given.

Lines 107-110 Corresponding clarifications for the preparation of PLA samples were made in Section 2. Materials and Methods.

All of the transition temperatures that are intrinsic to PLA (4% D) are listed in Section 2. Materials and Methods. In particular, the polymer can crystallize and melt, so the peak crystallization and melting temperatures are given as characteristics. However, we can obtain amorphous PLA samples under conditions of rapid cooling (quenching) of the melt. Samples with different structures have been obtained from such PLA. PLA-1 is a commercial film produced by extrusion followed by rapid cooling (quenching). It is transparent and, according to DSC data, has a degree of crystallinity close to 0% (the values of the heat of crystallization dHcryst = 22.9 J g-1 and the heat of melting dHmelt = 22.5 J g-1 in the DSC cell are almost equal). PLA-2 is a sample obtained by the rolling of PLA-1 film at room temperature 20-25 °C (well below the glass transition temperature of PLA). This process is accompanied only by orientation, but it does not lead to crystallization. The DSC data have confirmed that the PLA-2 sample has a degree of crystallinity close to 0% (the values of the heat of crystallization dHcryst = 28.9 J g-1 and the heat of melting dHmelt = 28.6 J g-1 in the DSC cell are almost equal).

4.

149     Were the composts and measured samples indeed prepared with the exactness to one ten-thousandth of weight as reported in Experimental.

Lines 133-144 We have edited the values to the nearest tenth.

5.

166.    Table 1.  Not a single unit in the column headed "Units" (which should be in the singular) is correct and/or complete.  

Table 1 has been modified.

Lines 152-153, Table 1 Corrected Electrical conductivity (EC) units from microsiemens to millisiemens. Removed units of quantities for pH and C/N. Clarified the units for Moisture content (wt.%).

6.

227     Authors could not calculate MW averages from PMMA standards. They determined apparent MW averages related to PMMA standards (See IUPAC Recommendations, Pure Appl. Chem. 2015; 87(1): 71–120, entry 2.13). This fact should be mentioned, but correctly, since overwhelming majority of published GPC MW data for synthetic polymers are based on PS standards.

We agree with the reviewer. The MWs of the polymer calculated from standards of another polymer are apparent.

Line 206 We have added “apparent” to in the corresponding sentence in Section 2.3.1.

7.

229     a) What is definition of “normalized weight fraction” ??    

b) Is the SEC coordinate dW/logM written correctly  ?? 

c) Note: the symbol w is commonly used for weight fraction.

d) Axis labels do not need to be in Experimental but should be in the figures. But they are missing there.

Lines 210-212 We have excluded this sentence from Section 2.3.1 and have labeled the axes on the corresponding Fig. 10 (Line 490) and Fig. 13 (Line 571).

8.

240     Determination of thermal stability.    Here, it is about a shape stability and not "chemical stability" usually understood under this heading.   Why not to state here that the side axis of ellipse was measured ??

We agree with the Reviewer. Indeed, the shape stability of PLA samples when heated was determined in the work. It was stated that “the side axis of ellipse was measured”.

Lines 221-222, 310 “thermal stability” has been replaced with “shape stability”

Line 228 It was stated that “the side axis of ellipse was measured”.

9.

254     CSR   useless abbreviation.

Line 235 The abbreviation CSR has been removed.

10.

257     FWHM  uspecified.

Lines 236-237 The abbreviation FWHM has been specified.

11.

271     A rather strange sentence.

Lines 250-251 The sentence has been changed.

12.

280     ANOVA   useless abbreviation.

Line 259 The abbreviation «ANOVA» has been removed.

13.

306     Caption to Tab. 2 does not correspond to its content. What chemical property is reported? Quantities not defined.

This is a printing error.

Line 286 The title of Table 2 has been corrected to read "Characteristics of PLA samples".

14.

318     The discussion to Figs 3 and 4 is very curious. These results show nothing more than the textbook knowledge of the effect of crystallinity on the basic mechanical properties of a crystallizable polymer and the shape memory of the deformed but not thermally-relaxed polymer. Discussion does not bring anything new.

Perhaps the results described are textbook knowledge (which is not surprising). However, they characterize the initial structurally different samples. We believe this is important for understanding the effect of temperature on the behavior of samples under composting condition.

15.

323     No crystalline but semi-crystalline films are studied.

We agree that semi-crystalline PLA samples were used.

The text (Lines 305, 317, 428, 452-456, 521 ) and Table 2 (Line 286) have been changed accordingly.

16.

348     This paragraph is essentially a repetition of facts from Introduction.

We agree with the Reviewer.

 Line 326 The paragraph has been changed.

17.

369     Text does not correspond with that to Fig. 5.

Line 337 Added  Fig. 5 “(dashed lines)” presents

18.

373     Info to Fig. 5 is absolutely insufficient.

Lines 342-345 There has been a correction to the caption of Fig. 5.

19.

425     Fig. 6.  Why low-crystallinity samples survive much longer than those oriented with higher degree of crystallinity?  Can you reasonably explain it ??? 

Fig. 6 shows the appearance of the samples after different periods of storage in standard compost. This Figure only shows that all types of samples lose their mechanical strength and fragment into pieces with increasing time. The loss of mechanical strength is associated not only with the processes of chemical degradation and the formation of defects in the form of pores and cracks, but also with crystallization under the conditions of the thermophilic phase of composting, which can occur differently for structurally different types of samples. In addition, sampling and cleaning are accompanied by mechanical stress, which can also lead to additional fragmentation. Therefore, it is incorrect to clearly correlate the size of fragments from PLA samples with the degradation rate. In addition, GPC studies showed (Fig. 10) that the change in Mn during the first week of composting (SC) was smallest for the oriented PLA-4 sample, which has the highest degree of crystallinity. This indicates the influence of the initial structure on the hydrolytic degradation rate, apparently associated with a decrease in water diffusion rate into the volume of oriented and semicrystalline samples. Furthermore, the degradation process of PLA can be accelerated by autocatalysis, which, as previously shown in [H. Tsuji, A. Mizuno, Y. Ikada, J. Appl. Polym. Sci. 2000. 77(7), 1452-1464], can occur faster in crystalline polymer due to the concentration of carboxyl groups formed during hydrolysis in the amorphous regions.

20.

475     Are data in Tab. 4 in accord with the DSC records in Fig. 8 ???

The data in Fig. 8 and Table 3 do not conflict and are consistent. Table 4 is not in the text of the manuscript.

Reviewer 2 Report

The article describes structural changes to polylactides and the impact of these changes on composting of the polymers. Considering the problems of plastic pollution, the article is relevant. However, several aspects need to be considered.

1.     According to the title, the structural modification of the polylactide needs to be described clearly. At present, it is modification in shapes.

2.     Abstract should be rewritten to provide some data (values). At present it is written in a very generalized way.

3.     There are many language errors that need to be addressed, pay special attention to avoid verb confusion.

4.     L47-48: Since it is not an opinion article, so, rephrase it.

5.     L107: Write in past tense. Also, revise to state the parameters on which the effects of structural and mechanical modifications were studied.

6.     2.1: Change the heading, instead of ‘object’ write ‘materials’.

7.     L145-148: Revise. Correct the verb. Modify to explain accelerated compost.

8.     L149-151: What was the rationale of using this composition? Any information on C:N ratio will bring more interest.

9.     L289: The same medium composition cannot be used to grow bacteria and fungi, both.

10.  L291-293: It is not clear if colony PCR was performed or whole genome was extracted.

There are several sentences with verb confusion and wrong choice of verbs.

Author Response

Dear Editor! We are grateful to the Reviewer for carefully reading our paper and for the valuable comments and suggestions that have helped to improve the manuscript. We agree with all the comments. We have tried to improve the manuscript significantly and hope that it now meets all requirements. The text of the manuscript has been extensively revised, so only those places where the Reviewer had comments are highlighted in yellow. Best regards, the authors.

 No.

Reviewer Comments

Response to Reviewer 2

1.

According to the title, the structural modification of the polylactide needs to be described clearly. At present, it is modification in shapes.

In this work, 4 types of PLA (4% D) film samples were studied, differing in structure, namely orientation and crystallinity: isotropic amorphous PLA-1; oriented amorphous PLA-2; isotropic semi-crystalline PLA-3; oriented semi-crystalline PLA-4.   The methods for obtaining samples of each type are described in sufficient detail in 2.1 Materials (experimental part); Table 2 shows their characteristics. Orientation and crystallization change the structure of the polymer, not its shape.

Lines 107-110 Sample preparation methods have been clarified.

2.

Abstract should be rewritten to provide some data (values). At present it is written in a very generalized way.

Lines 13-20  The abstract has been rewritten.

3.

There are many language errors that need to be addressed, pay special attention to avoid verb confusion.

We have tried to correct language errors throughout the manuscript.

4.

L47-48: Since it is not an opinion article, so, rephrase it.

Lines 39-41 The sentence has been rephrased.

5.

L107: Write in past tense. Also, revise to state the parameters on which the effects of structural and mechanical modifications were studied.

Lines 92-98 The aim of the study has been clarified. Sentences have been rewritten in the past tense.

6.

2.1: Change the heading, instead of ‘object’ write ‘materials’.

Line 100 Title 2.1 has been corrected to "Materials".

7.

L145-148: Revise. Correct the verb. Modify to explain accelerated compost.

Lines 133-135 Moved the sentence to lines 149-150.
Lines 133-134
Added clarification for "accelerated compost".

8.

L149-151: What was the rationale of using this composition? Any information on C:N ratio will bring more interest.

Lines 136-137 We've added the information: "The FW consisting of expired products of characteristic composition is obtained at the waste processing facility (Moscow region, coordinates: 56.048547, 36.996277)".

9.

L289: The same medium composition cannot be used to grow bacteria and fungi, both.

Lines 269-273 This point is really important for understanding the methodology used, we have added this information to the work.

It should be noted that when isolating a culture of the micromycete, an antibiotic was added to the medium (100 mg l-1 tetracycline). Thus, selective conditions were created, and normal development of fungi was achieved. In general, the medium of this composition was chosen because of its universality and its ability to provide growth of both bacteria and fungi. Both the commonly used LB (Luria-Bertani) medium for bacteria and Sabouraud's medium for fungi have a similar composition. In addition, media more commonly used for bacteria, including the aforementioned LB [Ishii, N., Inoue, Y., Tagaya, T., Mitomo, H., Nagai, D., & Kasuya, K. Polymer Degradation and Stability. 2008. 93(5), 883–888. doi:10.1016/j.polymdegradstab.200], are sometimes used for the cultivation of fungi. Therefore, to obtain an enrichment culture, we considered it possible to use the medium described in this work.

10.

L291-293: It is not clear if colony PCR was performed or whole genome was extracted.

Lines 271-275 We have clarified the presentation of this information. Initially, enrichment cultures were obtained on the described medium. Subsequently, pure cultures of microorganisms (2 bacterial and 1 micromycete culture) were isolated on the same compositional medium and further identified by 16S rRNA gene region for bacteria and ITS region for fungi. Whole genome extraction was not performed in this study.

11.

Moderate editing of English language required.

Comments on the Quality of English Language.

There are several sentences with verb confusion and wrong choice of verbs.

We performed English language editing of the entire text by a native English speaker after making corrections based on the comments of Reviewers.

Round 2

Reviewer 2 Report

Authors have acommodated the comments/suggestions given on the original draft. 

Some changes need to include as 

L16: Delete ‘number’

L17-18: State the name of the inoculant

Some minor changes particularly regarding the choice of words are needed. 

Author Response

Dear Editor! We have re-edited the English language in the manuscript. Changes in the manuscript text are marked in blue. Below you will find responses to the Reviewer's comments. Best regards, the authors.

Authors have accommodated the comments/suggestions given on the original draft.

Some changes need to include as 

 L16: Delete ‘number’

Response:

We recommend not to delete the word 'number' because there are several types of average molecular weights for polymers, i.e. Mn, Mw, Mz. In the abstract of the article, it is the number-average molecular weight of Mn that is used. Line 16: Added '(Mn)' to the sentence. 

L17-18: State the name of the inoculant

Response:

Line 18: Replaced 'inoculum' with 'inoculant'.

Comments on the Quality of English Language

Some minor changes particularly regarding the choice of words are needed. 

Response:

We have re-edited the English language in the manuscript. Changes in the manuscript text are marked in blue.